# Role of Rapid Antigen Testing in Population-Based SARS-CoV-2 Screening

**DOI:** 10.3390/jcm10173854

**Published:** 2021-08-27

**Authors:** Vicente Martín-Sánchez, Tania Fernández-Villa, Ana Carvajal Urueña, Ana Rivero Rodríguez, Sofía Reguero Celada, Gloria Sánchez Antolín, José Pedro Fernández-Vázquez

**Affiliations:** 1Research Group on Gene-Environment Interactions and Health (GIIGAS), Institute of Biomedicine (IBIOMED), Universidad de León, 24071 León, Spain; vmars@unileon.es; 2Consortium for Biomedical Research in Epidemiology & Public Health (CIBER en Epidemiología y Salud Pública—CIBERESP), 28029 Madrid, Spain; 3Animal Health Department, Universidad de León, 24071 León, Spain; amcaru@unileon.es; 4Gerencia de Atención Primaria, 24008 León, Spain; ariveror@saludcastillayleon.es (A.R.R.); sreguero@saludcastillayleon.es (S.R.C.); jpfernandez@saludcastillayleon.es (J.P.F.-V.); 5Castilla y Leon Health Department, Universidad de Valladolid, 47002 Valladolid, Spain; gsanchezan@saludcastillayleon.es

**Keywords:** SARS-CoV-2, screening, antigen

## Abstract

This study evaluates a population-based screening of asymptomatic people, using a rapid antigen diagnostic test (RADT), in areas of high transmission. To detect sources of SARS-CoV-2 infection, nasopharyngeal samples were taken and were tested using RADT. Confirmatory RT-qPCR tests were performed in both positive and negative cases. The internal validity of the RADT, the prevalence of infection, and the positive and negative predictive values (PPV and NPV) were estimated, based on the percentages of confirmed cases with 95% confidence interval. Of the 157,920 people registered, 50,492 participated in the screening; 50,052 were negative, and 440 were positive on the RADT (0.87%). A total of 221 positive RADT samples were reanalysed using RT-qPCR and 214 were confirmed as positive (96.8%; 95% CI: 93.5–98.7%), while 657 out of 660 negative RADT samples were confirmed as RT-qPCR negative (99.5%; 95% CI 98.7–99.9%). The sensitivity obtained was 65.1% (38.4–90.2%) and the specificity was 99.97% (99.94–99.99%). The prevalence of infection was 1.30% (0.95–2.13%). The PPVs were 95.4% (85.9–98.9%) and 97.9% (93.3–99.5%), respectively, while the NPVs were 99.7% (99.4–100%) and 99.2% (98.7–100%), respectively. The high specificity found allow us to report a high screening performance in asymptomatic patients, even in areas where the prevalence of infection was less than 2%.

## 1. Introduction

The World Health Organization (WHO)’s overall objective in their strategy against COVID-19 is for all countries to control the pandemic by stopping transmission and reducing the mortality associated with it. One of the key aspects in the prevention of transmission is the active search for and isolation of infected patients and the tracking and quarantining of their contacts, keeping in mind the high percentage of asymptomatic patients and the possibility of transmission before the onset of symptoms [1]. The screening strategy in the asymptomatic population is controversial and its efficacy is not well-established. However, some models estimate that to control SARS-CoV-2 transmission effectively, it is essential to reduce the risk of transmission by infected people who do not have symptoms [2].

The Spanish Ministry of Health suggests that in geographic areas and/or populations with high transmission rates and limited PCR (Polymerase Chain Reaction) resources, screening through rapid antigen detection tests (RADTs) could be useful [3]. There are several rapid commercial tests to detect SARS-CoV-2 antigens, but more experience is needed to know their internal validity, especially in screening contexts [4]. In a population screening with a low pre-test probability, it is of special interest to know the specificity of the test since it is the main determinant of the positive predictive value (PPV) [5]. The use of RADT in asymptomatic individuals in a context of low prevalence of SARS-CoV-2 infection could result in an unacceptably low PPV, and confirmation tests could be essential in the case of positives [6].

For these reasons, the purpose of this research was to carry out a preliminary evaluation of a RADT by determining its internal validity in the context of a population screening to detect the sources of infection by SARS-CoV-2.

## 2. Materials and Methods

### 2.1. Design

A descriptive observational study was carried out in eight basic health areas of León province (Castile and Leon, Spain): the basic health area of Cistierna (CBHA), the basic health area of Santa María del Páramo (SMPBHA), and the six basic health areas of the city of León and its conurbation (LBHAs).

Population-based screening was conducted when the 7-day cumulative incidence was above 500 cases per 100,000 members of the population and rising, and traceability was less than 70%, indicating community transmission. The COVID-19 incidence was calculated by taking into account the number of patients with an active infection diagnostic test, including RT-qPCR, RADTs or high-performance serological tests (ELISA, CLIA, ECLIA), in relation to the number of people registered in health care system, coverage of more than 95% of the population.. This epidemiological information came from the National Epidemiological Surveillance Network.

The population screening took place on 6 January 2021 in CBHA, on 13 January 2021 in SMPBHA, and on six consecutive days, between 3 and 8 February 2021, in LBHAs. A few days before the screening, the seven-day cumulative incidence rates per 100,000 population in the areas studied were 1203 in CBHA, 894 in SMPBHA, and 956 in LBHAs. The evolution of the seven-day cumulative incidence rates in the areas studied and the screening dates are shown in Figure 1 [7].

### 2.2. Population

The study population included a total of 157,920 people registered within the BHAs. The female-male ratio was 1.14 and a quarter of the population were aged 65 years or over. (Table 1).

### 2.3. Detection of SARS-CoV-2 Sources of Infection

In order to detect sources of SARS-CoV-2 infection, nasopharyngeal samples were taken with a swab and were tested using the SARS-CoV-2 Rapid Antigen Test (Roche, Switzerland). This test is a qualitative membrane-based immunoassay (immunochromatography) for the detection of the nucleocapsid (N) protein of SARS-CoV-2 in nasopharyngeal samples. When invalid results occurred, the performance of a new test was urged. All positive cases in CBHA, SMPBHA, and LBHAs were invited to give a new sample of nasopharyngeal exudate on the same day or the day after, which was resuspended in a commercial viral transport medium (VTM, 3 mL + Nasopharyngeal flocked swab, Deltalab Ref: 304295), and submitted for a RT-qPCR test on the same day. In one of the sample obtainment lines in the LBHAs screening, leftover samples (a nasopharyngeal swab inserted into an extraction buffer) were resuspended in VTM and used to perform RT-qPCR in negative cases. The RNA extraction was performed using a commercial kit (QIAamp cador Pathogen Mini Kit, Indical Bioscience, Leipzig, Germany), according to the protocol provided by the manufacturer and starting from 200 µL of VTM, and qRT-PCR was carried out with a QuantStudio 5 (Applied BioSystems^TM^, California, USA) using a commercial kit (BGI Genomics, Shenzhen, China), targeting the ORF1ab gene of SARS-CoV-2.

The sampling, reading, and interpretation of the results were carried out by trained personnel. Tests showing weak lines were considered positive. In doubtful cases, patients were invited to repeat the test by submitting a new sample. All positive cases were isolated and referred to their primary care physician, and contact tracing was started on the same day.

### 2.4. Statistical Analysis

Access to the screening program was evaluated by the proportion of screened subjects to the total number of people surveyed, stratifying by sex and age group.

The percentage of individuals who tested positive on the RADT was calculated by taking into account the total number of people who accessed the screening and stratifying by age group and sex; the 95% CIs were calculated for a Poisson distribution [8].

The observed percentages of true positive and true negative values with 95% confidence intervals were calculated using RT-qPCR as the gold standard, and were extrapolated to the total RADT screening results. In this way, the internal validity of the RADTs and the prevalence of infection were calculated for three scenarios (observed results and the upper and lower values of 95% CI). The results obtained for the prevalence of infection and the internal validity of the RADTs in the three scenarios were used to estimate the negative and positive predictive values [9].

### 2.5. Ethical Considerations

Participation was entirely voluntary and the study protocol was approved by the Ethics Committee of the health area of León and the Bierzo (reference: 2126).

## 3. Results

In the health areas assessed, there were 157,920 people registered, 53.3% of them women and 65.1% of them over 39 years of age. Overall participation in the screening was 32.0%, ranging from 41.5% in SMPBHA to 31.2% in LHBAs. In all areas and in the pooled analysis there was a higher participation of women than men (33.5% vs. 30.3%), as well as of people between 40 and 64 years of age (38.8%) (Table 1). The percentage of positive RADTs was 0.87% (95% CI: 0.79–0.96%), ranging from 3.31% (95% CI: 2.72–3.98%) in SMPBHA to 0.61% (95% CI: 0.54–0.69%) in LHBAs (Table 1). The prevalence observed was higher in men than in women, with significant differences only in the pooled analysis (men: 1.07%; 95% CI: 0.94–1.22%; women: 0.71%; 95% CI: 0.62–0.82%) (Table 1). The highest prevalence of positive RADTs was observed in the 15–39 age group, which showed statistically significant differences with those aged 65 and over (1.09%; 95% CI: 0.91–1.29% vs. 0.72%; 95% CI: 0.57–0.89%) (Table 1).

Of the 440 participants who tested positive by RADT, 214 were re-confirmed by RT-qPCR and 7 yielded a negative result, while 219 did not agree to have a new sample taken for confirmation. The percentage of positive tests that were analyzed and confirmed with RT-qPCR was 96.8% (95% CI: 93.6–98.7%) (Table 2). The mean ((SD; Range)) number of cycles in the positive cases that underwent confirmation by RT-qPCR was (21.3; 0.29; 12.3–36.0). On the other hand, out of 660 negative cases re-analyzed with RT-qPCR, only 3 were identified as positive, with a cycle threshold (Ct) between 35 and 36. A negative result was confirmed in 657 cases (99.5%; 95% CI: 98.7–99.9%).

Extrapolating from the results obtained from the RT-qPCR confirmatory tests, the estimated prevalence of infection and the internal validity of the RADTs were estimated (Table 3). The estimated sensitivity was 65.14% (CI 95%: 38.36–90.23%) and specificity was 99.97% (CI 95%: 99.94–99.99%), while the prevalence of infection was 1.30% (CI 95%: 0.95–2.13%).

Figure 2 shows the resulting predictive values for the prevalence of infection and the estimated internal validity. Even in cases of low prevalence of infection, around 1%, the positive predictive value was higher than 95% and increased in value as the pre-test probability increased. Similarly, the negative predictive value was greater than 99% in all scenarios.

## 4. Discussion

The use of mass screening for the early detection and isolation of asymptomatic sources of infection of SARS-CoV-2 is controversial, especially when it involves the use of RADT. The WHO recommends screening using RADT only in cases of community transmission and in contexts where the pre-test probability is greater than 5%, such as in health facilities, COVID-19 testing centers/sites, nursing homes, prisons, schools, front-line and health-care workers, and for contact tracing [10]. According to several authors, the use of RADT in asymptomatic individuals in a context of low prevalence of SARS-CoV-2 infection will result in an unacceptably low positive predictive value (PPV), and confirmation tests are essential in the case of positives [6,11,12].

The performance of screening tests is defined by the pre-test probability and the internal validity of the test used. In the investigated areas, the pre-test probability were low even for a pandemic situation and significant community transmission, because of SARS-CoV-2 is an acute infec-tion, with an average duration of viral shedding of around 10 days (two days before and one week after the onset of the symptoms) in mild or asymptomatic cases [13]. Indeed, as we observed in this study, even in the most unfavorable scenario the prevalence of sources of infection would be 2%; it is most likely to be around 1%. In any case, prevalence would be below the 5% at which the WHO recommend screening with RADT. We are not aware of any published population-based screening studies with observed prevalence above 5% except for in Slovakia, where a total of 140,951 individuals were tested and 5594 positive cases were identified (3.97%), and real prevalence was probably higher than 5% [14].

In this low-prevalence scenario, the main determining factor of the screening test’s performance is its specificity. If (1-specificity) is higher than the pre-test probability, then the vast majority of positives will be false positives, thereby undermining the screening test’s performance [1,9]. However, the specificity found in this screening program, of 99.97% or, in the worst-case scenario, of 99.94%, in a context of low pre-test probability, resulted in a very acceptable performance. Almost all the positive cases that were detected were sources of infection of SARS-CoV-2. Thus, the positive predictive value was above 85% in all scenarios and close to 100% in many of them.

According to the manufacturer, the sensitivity of the test used in the screening is 96.52%, and the specificity is 99.68%. Several published studies regarding the internal validity of the Roche RADT suggest that it has a high level of sensitivity in symptomatic patients with a low Ct and at the onset of the symptomatic phase (100–76.2%), which decreases in the case of a high Ct (22.0–49.0%) and also in asymptomatic patients. However, these same authors also find very high specificities, sometimes up to 100%, in asymptomatic patients (96.0–100%) [15,16,17,18]. These data are consistent with the results obtained by other authors in studies using different RADTs, where the observed specificity ranged between 95% and 100% and the sensitivity between 45% and 100%, depending on the presence or absence of symptoms, the time elapsed since the infection, and the average cycle threshold or Ct [19,20,21,22].

It is especially noteworthy that the Cts found in our screening among asymptomatic patients (21.3) were even lower than those reported by other authors in symptomatic patients (23.3) [20,21]. The fact that asymptomatic patients may present viral loads similar to symptomatic patients has been reported previously [22,23,24,25].

In our case, the disadvantage attributed by other authors to RADTs, of not having adequate specificity in a low-prevalence context, was not observed [20,21,22,23]. On the contrary, it was very useful to quickly and accurately detect individuals who were capable of transmitting the infection to others [22,23,25,26]

Another relevant issue is the high proportion of PCR-confirmed negative cases in antigen-negative participants, and the fact that three non-confirmed cases were found in subjects with very a high Ct, above 35. Even in the case of low sensitivity, it could be assumed that false negatives, infected subjects not detected by RADT, are cases with low viral loads that are therefore limited as potential sources of infection, at least in population screening and outside enclosed places, where there is a high potential for transmission (i.e., health facilities, COVID-19 testing centers/sites, care homes, prisons or schools) [25,26,27,28,29].

The sensitivity observed in our study is in line with that reported by other authors and may be considered low and insufficient as a reference method for the diagnosis of infection in asymptomatic patients [30]. However, it may be sufficient in a context of high incidence and community transmission, and where there are insufficient resources for mass RT-PCR testing, with the caveat that a negative test does not rule out the presence of infection.

Another advantage of the screening we carried out is that it was organized by the Primary Care Services and, therefore, all the activities of isolating and providing care for the patients and the tracing, quarantining and care of close contacts was integrated [31]. Furthermore, as stated by Mina et al. [32], it is not only the internal validity of a single diagnostic test which must be assessed. The context of its use and assessment within a Swiss cheese strategy should also be taken into account when setting pre-test probabilities and rethinking the pre-test probability level of 5% as a benchmark for the implementation of population screening strategies in asymptomatic people.

The prevalence of infection observed is in line with the cumulative incidences reported by health authorities and their epidemiological surveillance services. The best screening performances were observed in CBHA and SMPBHA, where cumulative incidence rates were higher and rising, although screening itself increased them markedly. In LBHAs, the rates were lower and declining, which may explain the lower yield and the lack of increase in incidences after screening, as was the case in the other areas. In all the models that analyzed the effectiveness of screening, we used the basic reproduction number or R0, which estimates the speed at which a disease is capable of spreading in a population [33].

The level of participation in the screening, at around one third of the registered population, can be considered high given its voluntary nature and that the consequence of the detection of a positive case was isolation. In this respect, the responsibility and solidarity of the participants in the screening should be highlighted. It should also be noted that, as expected, participation was higher in places with a higher incidence and, therefore, higher perceived risk; in young adults, for the same reason; and in women, who tend to participate in screening studies to a greater extent than men. Similarly, the prevalence of detected infection was lower in women and in the 65+ age group. Women’s lower risk-taking and greater adherence to prevention recommendations may explain these results, while the special vulnerability of older people and their perception of risk may lead them to adopt effective prevention measures more frequently [34].

There are few published studies on real-life mass screening using RADT, and this is one of the main strengths of our study. However, the voluntary nature of our screening and the associated selection bias should lead us to be cautious and to bear this in mind when interpreting the results, although these results are very likely to underestimate the sources of infection and therefore the performance of the screening. We must not forget the law of inverse prevention [35]. Another limitation of our study was the fact that only half of the positive samples were re-tested and confirmed on a molecular platform. Furthermore, the number of samples analyzed was not high, therefore the influence of chance on the results obtained must be taken into account. However, similar results to ours have been published by other authors regarding the sensitivity and specificity of RADTs.

## 5. Conclusions

This analysis found a high specificity for the RADT in a real-life mass population screening, which resulted in a high efficiency rate even in the context of a prevalence of infected people below 5%. Very high predictive values were observed which, if confirmed, may suggest that confirmatory testing of antigen test positives could be unnecessary. The limitations associated with the low RADT sensitivity were compensated for by a quick, cheap, and accessible diagnostic method. The participation of the population, on a voluntary basis, was acceptable and performance improved where incidence rates were high and clearly rising. Given the high percentage of asymptomatic sources of infection, prevention and control of SARS-CoV-2 infection should incorporate active search strategies for asymptomatic individuals. Although our results should be interpreted with caution, these data open the door to new diagnostic strategies using the RADT in order to decrease the incidence of SARS-CoV-2.

## Figures and Tables

**Figure 1 jcm-10-03854-f001:**
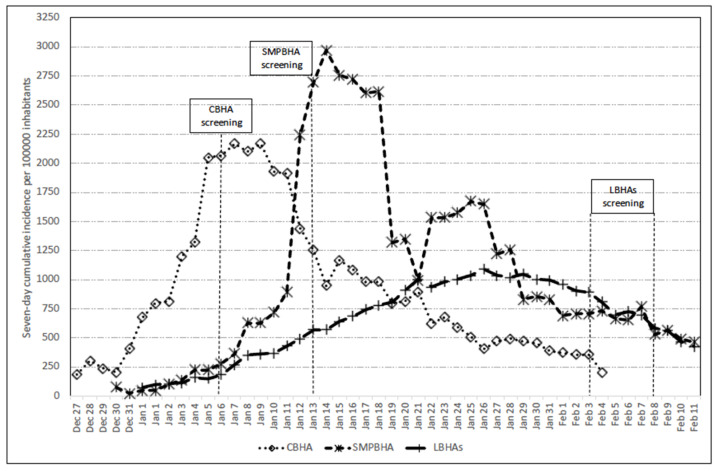
Evolution of the seven-day cumulative incidence rates in the basic health areas studied. The dates of population screening within each area are indicated.

**Figure 2 jcm-10-03854-f002:**
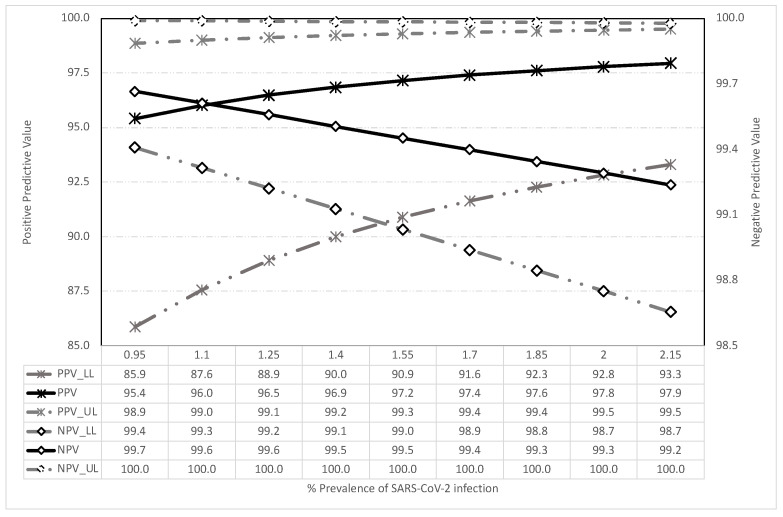
Predictive values obtained according to the prevalence of SARS-CoV-2 infection: LL = Lower limit; UL = Upper limit.

**Table 1 jcm-10-03854-t001:** Population distribution among registered, screened, RADT-positive individuals, and SARS-CoV-2 infection prevalence detected by RADT by age and sex.

Health Area/Variables			Registered	Screened		Positive RADTs			
			N	n	%	n	%	95% CI	
CBHA			5904	2281	38.6	55	2.41	1.82	3.13
	Age (years)								
		0–14	418	108	25.8	1	0.93	0.23	5.05
		15–39	1064	544	51.1	21	3.86	2.41	5.84
		40–64	2424	1120	46.2	26	2.32	1.52	3.38
		≥ 65	1998	509	25.5	7	1.38	0.55	2.81
	Sex								
		Men	3052	1157	37.9	32	2.77	1.90	3.88
		Women	2852	1124	39.4	23	2.05	1.30	3.05
SMPBHA			7943	3296	41.5	109	3.31	2.72	3.98
	Age (years)								
		0–14	547	164	30.0	3	1.83	0.38	5.25
		15–39	1626	660	40.6	28	4.24	2.84	6.07
		40–64	2894	1310	45.3	47	3.59	2.65	4.74
		≥ 65	2979	1162	39.0	31	2.67	1.82	3.77
	Sex								
		Men	4054	1619	39.9	68	4.20	3.28	5.29
		Women	3889	1677	43.1	41	2.44	1.76	3.30
LBHA			144,073	44,915	31.2	276	0.61	0.54	0.69
	Age (years)								
		0–14	16,356	3142	19.2	31	0.99	0.67	1.40
		15–39	35,200	10,289	29.2	76	0.74	0.58	0.92
		40–64	56,573	21,573	38.1	124	0.57	0.48	0.68
		≥ 65	35,944	9911	27.6	45	0.45	0.33	0.61
	Sex								
		Men	66,603	19,534	29.3	139	0.71	0.60	0.84
		Women	77,470	25,381	32.8	137	0.54	0.45	0.64
Pooled			157,920	50,492	32.0	440	0.87	0.79	0.96
	Age (years)								
		0–14	17,321	3414	19.7	35	1.03	0.72	1.42
		15–39	37,890	11,493	30.3	125	1.09	0.91	1.29
		40–64	61,891	24,003	38.8	197	0.82	0.71	0.94
		≥ 65	40,921	11,582	28.3	83	0.72	0.57	0.89
	Sex								
		Men	73,709	22,310	30.3	239	1.07	0.94	1.22
		Women	84,211	28,182	33.5	201	0.71	0.62	0.82

**Table 2 jcm-10-03854-t002:** Results of samples retested with RT-qPCR.

RADT	Basic Health Area	Total	RT-qPCR					
			N (%)	Positive	Negative	%	95% CI	
Positive	CBHA	55	51 (92.7%)	50	1	98.0 ^1^	89.6	100.0
	SMPBHA	109	102 (93.6%)	101	1	99.0 ^1^	94.7	100.0
	LBHAs	276	68 (24.6%)	63	5	92.6 ^1^	83.7	97.6
	POOLED	440	221 (50.2%)	214	7	96.8 ^1^	93.6	98.7
Negative	LBHAs	44639	660 (1.5%)	3	657	99.5 ^2^	98.7	99.9

^1^ Percentage of true positives. ^2^ Percentage of true negatives.

**Table 3 jcm-10-03854-t003:** Internal validity of the RADTs according to the RT-qPCR results.

Scenarios			RADT		Total	Sensitivity (%)	Specificity (%)	Prevalence of Infection %
			Positive	Negative				
Total			440	50,052	50,492			
Upper limit of the 95% CI	RT-qPCR	Positive	434	47	481	90.23	99.99	0.95
		Negative	6	50,005	50,011			
Observed	RT-qPCR	Positive	426	228	654	65.14	99.97	1.30
		Negative	14	49,824	49,838			
Lower limit of the 95% CI	RT-qPCR	Positive	412	662	1074	38.36	99.94	2.13
		Negative	28	49,390	49,418			

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
