# Peer review of "Role of Rapid Antigen Testing in Population-Based SARS-CoV-2 Screening"

_jcm, 2021, doi:10.3390/jcm10173854_

Round 1

Reviewer 1 Report

The authors describe the large-scale use of RADT in an a priori asymptomatic population and in a context of low COVID-19 prevalence. Although this topic has already been addressed in the literature, the text is well written. However, some points need clarification.

Line 51: The authors emphasise the importance of specificity. However, if the goal is screening asymptomatic people in order to allow them to travel or carry out activities while avoiding the spread of the virus, the authors should focus more on sensitivity.

Line 61: what are the screening indications for the population studied: before a trip abroad? before admission to hospital? Was the time of the screening chosen randomly or deliberately at a given time? For CBHA and SMPBHA (figure 1), it seems that the authors waited for a peak to start the screening.

Line 91: « gene » instead of « gen »

Line 91: it would have been interesting to use a molecular method also detecting the N gene

Line 82: qualitative membrane-based immunoassays are sometimes difficult to read since the line of tests is sometimes very weak. What was the authors' policy on this subject? The reading method should be more explicit: automated and standardized reading? If visual reading, did a second operator review the tests? Was a barely visible line considered positive?

Table 1: 144,073 instead of 14,4073

Table 2: « RADT » instead of « RDTA »

Line 132: On what criteria were the positive and negative samples selected for being retested by RT-PCR?

Line 147: where does the 65.14% sensitivity come from? It is difficult to understand : 426/654 true positives from Table 3, while (line 132 and Table 2) 214 RADT positives were tested in PCR (214/221=0.968).

Figure 2: What does _Ls and _Li mean? Please define the abbreviations in the legend.

Line 205: Please report the Ct observed in Results as well. The authors did not analyse the impact of the cycle threshold on the observed RADT performance. To be implemented in Results.

Author Response

According to the notification received, in which changes are proposed to the article entitled " Role of rapid antigen testing in population-based SARS-CoV-2 screening" (Manuscript ID: jcm-1325775), the authors note below in black the e-mail received and in blue the changes made.

The authors indicate that changes made to the main text of the manuscript are marked in red to facilitate identification.

Comments from reviewer 1

The authors describe the large-scale use of RADT in an a priori asymptomatic population and in a context of low COVID-19 prevalence. Although this topic has already been addressed in the literature, the text is well written. However, some points need clarification.

Line 51: The authors emphasise the importance of specificity. However, if the goal is screening asymptomatic people in order to allow them to travel or carry out activities while avoiding the spread of the virus, the authors should focus more on sensitivity.

We agree with the reviewer's statement. In the case of screening for travel or similar situations, sensitivity is important.  However, the context of our article is different, it is a mass population screening for sources of infection in a community transmission situation and without sufficient resources to perform PCR. In this context, there are authors who state the need for confirmatory PCR testing in the case of positives due to insufficient specificity. In our case, what is important is specificity in deciding whether to perform confirmatory RT-PCR in the case of antigen test positives. In our study we observed a high specificity and therefore a high positive predictive value that does not justify the recommendation of confirmatory RT-PCR in positive cases.

Line 61: what are the screening indications for the population studied: before a trip abroad? before admission to hospital? Was the time of the screening chosen randomly or deliberately at a given time? For CBHA and SMPBHA (figure 1), it seems that the authors waited for a peak to start the screening.

In this case, there were several mass screenings in which the entire population of specific geographical areas was invited to participate. The criteria for deciding when and where to screen were based on 7-day cumulative incidences and traceability in the search for contacts.  We have added these criteria in material and methods.

“Population-based screening was conducted when the 7-day cumulative incidence was above 500 cases per 100,000 population and rising and traceability was less than 70% indicating community transmission.”

Line 91: « gene » instead of « gen »

Changed.

Line 91: it would have been interesting to use a molecular method also detecting the N gene

Although the target of the SARS-CoV-2 Rapid Antigen Test used in this research is the nucleocapsid or N protein of SARS-CoV-2, we do not consider necessary to use the same viral target for the molecular test used. The molecular method (RT-qPCR) used has been authorized by the U.S. Food and Drug Administration (FDA) and has a detection limit similar to other commercial qPCR tests for SARS-CoV-2 detection (100 copies/mL). Moreover, its target within the ORF1ab gene is a highly conserved region among different strains of SARS-CoV-2 and it is routinely used for SARS-CoV-2 infection detection by health services in Spain.

Line 82: qualitative membrane-based immunoassays are sometimes difficult to read since the line of tests is sometimes very weak. What was the authors' policy on this subject? The reading method should be more explicit: automated and standardized reading? If visual reading, did a second operator review the tests? Was a barely visible line considered positive?

We add more information about this in section 2.3 of the manuscript.

"Sampling, reading of the result and its interpretation were carried out by trained personnel. Test with weak lines were considered as a positive case. In doubtful cases, the patient was invited to repeat the test by taking a new sample".

Table 1: 144,073 instead of 14,4073

Changed.

Table 2: « RADT » instead of « RDTA »

Changed.

Line 132: On what criteria were the positive and negative samples selected for being retested by RT-PCR?.

All positive cases were invited to take a new nasopharyngeal sample for confirmation by RT-PCR. Sampling was not performed when patients did not agree to it. Added in material and methods.

In the case of the negative results, only in the case of the León city screening, it was decided to carry out a reanalysis in one of the screening lines. As mentioned in the material and methods.

Line 147: where does the 65.14% sensitivity come from? It is difficult to understand : 426/654 true positives from Table 3, while (line 132 and Table 2) 214 RADT positives were tested in PCR (214/221=0.968).

The extrapolation criteria for estimating sensitivity, specificity and prevalence of infection in the pooled analysis of screening is described in Material and Methods section, lines 114-120.

The observed percentages of true positive and true negative with their 95% confidence intervals were calculated using RT-qPCR as the gold standard and were extrapolated to the total RADT screening results. In this way, the internal validity of the RADTs and the prevalence of the infection were calculated for three scenarios (observed results and the upper and lower values of 95% CI). The results obtained for prevalence of infection and internal validity of the RADTs in the three scenarios were used to estimate the negative and positive predictive values”.

Figure 2: What does _Ls and _Li mean? Please define the abbreviations in the legend.

Thank you for your comment. Figure 2 has been modified, because the abbreviations were in English by mistake. They have been changed to LL - lower limit and UL-upper limit.

Line 205: Please report the Ct observed in Results as well. The authors did not analyse the impact of the cycle threshold on the observed RADT performance. To be implemented in Results.

We do not know if the reviewer had any problems reading the manuscript. The required information is analysed in the results section between lines 149-153 (lines 139-143 of original manuscript). We indicate below the text indicated:

“The mean ([SD; Range]) number of cycles in the positive cases that underwent confir-mation by RT-qPCR was [21.3; 0.29; 12.3-36.0]). On the other hand, out of 660 negative individuals re-analyzed with RT-qPCR, only 3 were identified as positive with cycle threshold (Ct) between 35 and 36. Negative result was confirmed in 657 individuals (99.5%; 95% CI: 98.7%-99.9%).”

Reviewer 2 Report

  1. Line 67: The authors should specify the methods through which the incidence was calculated (RT-PCR tests? number of hospital admissions? RAT screening?) or provide reference.
  2. Line 81 (...with a swab...): The Authors should specify the method of transport of the swabs, if applicable. Were the swabs analyzed directly (dry swab) or transported to the laboratory in a transport medium (wet swab)?
  3. Line 86: Retesting on RT-PCR for sensitivity calculation should be done on the same sample to avoid pre-analytical and analytical biases.
  4. Line 87 (... the leftover antigen swab...): Again, more details are needed to explain the method of retesting with RT-PCR: was leftover antigen taken from a transport medium in which the swab was transported? Was the swab soaked in a VTM after the antigen test? Quantities of VTM used for RT-PCR testing should also be provided for transparency and reproducibility purposes.
  5. Results: Even though the screening strategy was mentioned in the introduction and abstract, the authors should specify here weather the screened patients were having COVID-19 symptoms at the time of screening.
  6. Line 132: The fact that only half of the positive samples were re-tested and confirmed on a molecular platform should be mentioned among the limitations of the study.
  7. Line 137: How were these 660 individuals chosen to be re-analyzed and why did the authors limit the molecular confirmation of negative samples to 660? Also, as mentioned above, please provide the details of reuse of the leftover antigen swab, without which it is not possible to interpret the sensitivity results.
  8. Line 170-171: I would suggest to implement the references with the following articles: https://doi.org/10.3390/diagnostics11071211, https://doi.org/10.3390/microorganisms9040798
  9. Line 217: The author should discuss the fact that a sensitivity of 65% is largely not sufficient to be considered the reference method among asymptomatic patients, but only to be carefully used in specific clinical settings, stressing the fact that it should only be used when RT-PCR is not available. Data from a recent epidemiological study (https://doi.org/10.1016/j.scitotenv.2021.147483) demonstrated the possible onset of clusters of infections originating from patients with VLs below 105 copies/mL. This indirectly highlights the risk associated with a RADT screening with low sensitivity.
  10. Line 260-261: This statement is too general and dangerous and it should be modified and toned down to avoid inducing mistakes in policy making and public health strategies. With a specificity range (of RADT) varying from 95 to 100%, eliminating the RT-PCR confirmation after RAD testing would imply accepting up to 5% of false positive results, which would lead to unnecessary quarantine measures. These, not only would have a negative impact at a socio-economic level, but also (and more importantly) could lead to the isolation of the person together with other real positive individuals, which would likely end up in the contagion of the patient initially tested falsely positive.
  11. Based on the reduced sensitivity (NPV below 90%) compared to RT-PCR, mass screening should be limited to non-vulnerable subjects, who should continue to get tested through molecular and more sensitive methods, in order to early diagnose the infection and allowing close monitoring of the disease.

Author Response

According to the notification received, in which changes are proposed to the article entitled " Role of rapid antigen testing in population-based SARS-CoV-2 screening" (Manuscript ID: jcm-1325775), the authors note below in black the e-mail received and in blue the changes made.

The authors indicate that changes made to the main text of the manuscript are marked in red to facilitate identification.

Comments from reviewer 2

Line 67: The authors should specify the methods through which the incidence was calculated (RT-PCR tests? number of hospital admissions? RAT screening?) or provide reference.

We are included more information in the text.

“The COVID-19 incidence has been carried out taking into account the number of patients with an active infection diagnostic test, including RT-qPCR, RADTs or high performance serological tests (ELISA, CLIA, ECLIA) in relation to the number of sanitary cards per basic health area. This epidemiological information comes from the National Epidemiological Surveillance Network”.

Line 81 (...with a swab...): The Authors should specify the method of transport of the swabs, if applicable. Were the swabs analyzed directly (dry swab) or transported to the laboratory in a transport medium (wet swab)?.

As mentioned in Material and Methods section, all positive cases were invited to take a new nasopharyngeal sample for confirmation by RT-PCR. The sample was collected on the same day or the day after using swabs that were immediately included in commercial viral transport tubes as it is usually carried out for routinely diagnosis of SARS-CoV-2 infection. Samples were submitted to the laboratory and processed by RT-qPCR within the same day. Manuscript has been modified to provide more clear information.

“All positive cases in CBHA, SMPBHA and LBHAs were invited to take a new sample of nasopharyngeal exudate on the same day or the day after. Sample was resuspended in a commercial viral transport media (VTM, 3 ml + Nasopharyngeal flocked swab, Deltalab Ref: 304295) and submitted to perform a RT-qPCR test within the same day.”

Moreover, information regarding the collection of negative samples for retesting has also been improved.

“In the LBHAs screening, in one of the taking samples lines, the leftover sample (naso-pharyngeal swab into extraction buffer) was resuspended in VTM and used to perform RT-qPCR in negative cases.”

Line 86: Retesting on RT-PCR for sensitivity calculation should be done on the same sample to avoid pre-analytical and analytical biases.

We agree with the reviewer's assessment. We tried in all cases to repeat the sampling on the same day, which was not always possible depending on the patient's response. In the case of negatives, the same sample from the same day was used. In the case of the positives, the vast majority of cases the PCR was repeated on the same day and in the remaining cases on the following day. It is therefore not expected to affect the results obtained.

Line 87 (... the leftover antigen swab...): Again, more details are needed to explain the method of retesting with RT-PCR: was leftover antigen taken from a transport medium in which the swab was transported? Was the swab soaked in a VTM after the antigen test? Quantities of VTM used for RT-PCR testing should also be provided for transparency and reproducibility purposes.

As already mentioned, all positive cases were invited to take a new nasopharyngeal sample for confirmation by RT-PCR. The sample was collected using swabs that were immediately resuspended in commercial viral transport tubes and submitted to the laboratory in the same day. For negative samples, the leftover sample (naso-pharyngeal swab into extraction buffer) was resuspended in VTM and used to perform RT-qPCR.

Both RNA extraction and RT-qPCR were carried out commercial kits and according to the manufacturers. However, the volume of VTM used for RNA extraction has been included as suggested for transparency and reproducibility purposes.

“RNA extraction was performed using a commercial kit (QIAamp cador Pathogen Mini Kit, Indical Bioscience) according to the protocol provided by the manufacturer and starting from 200 µL of VTM and qRT-PCR was carried out on a QuantStudio 5 (Ap-plied BioSystemsTM) using a commercial kit (BGI Genomics) targeting ORF1ab gen of SARS-CoV-2.”

Results: Even though the screening strategy was mentioned in the introduction and abstract, the authors should specify here weather the screened patients were having COVID-19 symptoms at the time of screening.

The call was for asymptomatic patients who were neither in quarantine nor in isolation, but no such questions were asked at the time of sampling. This comment is included in material and methods and discussion.

Line 132: The fact that only half of the positive samples were re-tested and confirmed on a molecular platform should be mentioned among the limitations of the study.

Thank you four your comment. We are included this sentence in limitations.

Line 137: How were these 660 individuals chosen to be re-analyzed and why did the authors limit the molecular confirmation of negative samples to 660? Also, as mentioned above, please provide the details of reuse of the leftover antigen swab, without which it is not possible to interpret the sensitivity results.

In the case of the negative results, only in the case of the León city screening, it was decided to carry out a reanalysis in one of the screening lines. As mentioned in the material and methods. The reasons were limited staff and resources.

Line 170-171: I would suggest to implement the references with the following articles: https://doi.org/10.3390/diagnostics11071211

https://doi.org/10.3390/microorganisms9040798

 Thank you for your comment, we included these references in the manuscript.

Line 217: The author should discuss the fact that a sensitivity of 65% is largely not sufficient to be considered the reference method among asymptomatic patients, but only to be carefully used in specific clinical settings, stressing the fact that it should only be used when RT-PCR is not available. Data from a recent epidemiological study (https://doi.org/10.1016/j.scitotenv.2021.147483) demonstrated the possible onset of clusters of infections originating from patients with VLs below 105 copies/mL. This indirectly highlights the risk associated with a RADT screening with low sensitivity.

Thank you for this comment. We have included the citation in the document and modified the text with the justification.

“The sensitivity observed is in line with that reported by other authors and may be considered low and insufficient as a reference method for the diagnosis of infection in asymptomatic patients [30]. However, it may be sufficient in a context of high incidence, community transmission and insufficient resources for mass RT-PCR testing, with the caveat that a negative test does not rule out the presence of infection”.

Line 260-261: This statement is too general and dangerous and it should be modified and toned down to avoid inducing mistakes in policy making and public health strategies. With a specificity range (of RADT) varying from 95 to 100%, eliminating the RT-PCR confirmation after RAD testing would imply accepting up to 5% of false positive results, which would lead to unnecessary quarantine measures. These, not only would have a negative impact at a socio-economic level, but also (and more importantly) could lead to the isolation of the person together with other real positive individuals, which would likely end up in the contagion of the patient initially tested falsely positive.

Based on the reduced sensitivity (NPV below 90%) compared to RT-PCR, mass screening should be limited to non-vulnerable subjects, who should continue to get tested through molecular and more sensitive methods, in order to early diagnose the infection and allowing close monitoring of the disease.

The authors agree with the reviewer's comment. The text has been changed.

 “Very high predictive values have been observed which, if confirmed, may not make confirmatory testing of antigen test positives necessary”.

Round 2

Reviewer 1 Report

No comments anymore.